# SPARTAN: A Sparse Transformer World Model Attending to What Matters

**Anson Lei**
Applied AI Lab
University of Oxford, UK
anson@robots.ox.ac.uk

**Bernhard Schölkopf**
MPI for Intelligent Systems
Tübingen, Germany
bs@tue.mpg.de

**Ingmar Posner**
Applied AI Lab
University of Oxford, UK
ingmar@robots.ox.ac.uk

## Abstract

Capturing the interactions between entities in a structured way plays a central role in world models that flexibly adapt to changes in the environment. Recent works motivate the benefits of models that explicitly represent the structure of interactions and formulate the problem as discovering *local causal structures*. In this work, we demonstrate that reliably capturing these relationships in complex settings remains challenging. To remedy this shortcoming, we postulate that sparsity is a critical ingredient for the discovery of such local structures. To this end, we present the *SPARse TrANsformer World model* (SPARTAN), a Transformer-based world model that learns context-dependent interaction structures between entities in a scene. By applying sparsity regularisation on the attention patterns between object-factored tokens, SPARTAN learns sparse, context-dependent interaction graphs that accurately predict future object states. We further extend our model to adapt to sparse interventions with *unknown* targets in the dynamics of the environment. This results in a highly interpretable world model that can efficiently adapt to changes. Empirically, we evaluate SPARTAN against the current state-of-the-art in object-centric world models in observation-based environments and demonstrate that our model can learn local causal graphs that accurately reflect the underlying interactions between objects, achieving significantly improved few-shot adaptation to dynamics changes, as well as robustness against distractors.

## 1 Introduction

*World Models* [11] have emerged in recent years as a promising paradigm to enable a wide range of downstream tasks such as video prediction [50, 13], physical reasoning [7], and model-based RL [12]. Recent advances have employed the transformer architecture [48] to develop world models capable of performing accurate predictions over ever longer horizons in increasingly complex settings [30, 38]. Owing to the flexibility of attention mechanisms, the transformer architecture can be combined with object-centric representations [28] to accurately capture interactions between objects, achieving state-of-the-art prediction performance [53]. However, the ability to generalise and adapt to *changes* in the environment in a data-efficient manner remains a significant challenge. Evidence suggests that transformer world models can be sensitive to changes in distractors [39], further highlighting the need for models to capture robust and generalisable interactions. As such, recent works have

39th Conference on Neural Information Processing Systems (NeurIPS 2025).

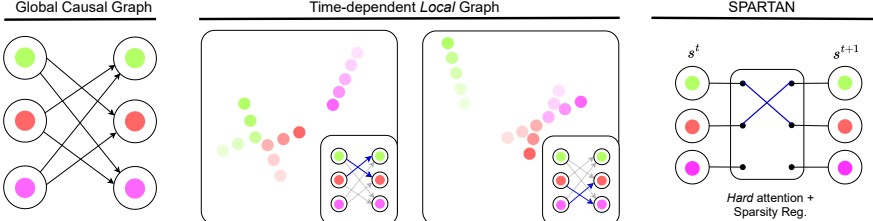

Figure 1: In the context of modelling physical interactions, a global causal graph is often uninformative and close to fully-connected. A time-dependent *local* causal graph better captures the sparse nature of interactions between entities. We present SPARTAN, a Transformer-based world model that discovers local causal structure using hard attention with sparsity regularisation.

argued that explicitly representing interaction structures between entities, i.e. *when* and *how* entities influence each other, is critical in improving generalisation [35] and adaptation efficiency [25, 14].

To this end, the intersection between causality and machine learning [40] offers a promising framework for building structured models that capture interactions in a principled manner. Through this lens, the forward dynamics can be factorised into *independent causal mechanisms*, where prediction for each entity is made based solely on other entities that have a causal influence on it. Here, the *Sparse Mechanism Shift* hypothesis [41, 4] posits that natural changes in the data distribution can be explained by *sparse* changes in these causal mechanisms. This suggests that a world model reflecting the causal structure of the world can adapt *efficiently* to changes by updating only a small part of the model while most of the model remains invariant. This causal view of world modelling has motivated prior works which explored the benefits of structured world models [25, 14]. These approaches aim to learn a graph that encodes how entities in an environment influence each other. However, these methods learn a fixed graph to fit an entire dataset, which is unsuitable for modelling realistic physical interactions, such as collisions between objects, that manifest as temporally sparse events. To remedy this, recent approaches [42, 15] formulate the problem as learning *state-dependent* local causal graphs (Fig. 1). Despite their success in simple state-based settings, scaling prior local causal discovery methods to image observations with complex dynamics remains a significant bottleneck. This hinders the application of these methods to realistic scenarios where, for example, objects are represented via learned embeddings and the number of objects can vary from scene to scene.

Taking inspiration from the conceptual framework of local causal models, the goal of this work is to develop a transformer-based world model that faithfully reflects the interaction structure between objects while maintaining the flexibility and accuracy afforded by the transformer architecture. The key insight is that attention mechanisms serve as a natural starting point for constructing state-dependent graphs. By applying sparsity regularisation, an inductive bias used in prior local causal discovery methods [15], on a modified transformer architecture with *hard* attention, we develop SPARTAN, a Transformer-based world model with a learnable local causal graph. We evaluate our model on observation-based environments with physical interactions and a real-life traffic motion prediction dataset [44]. To the best of our knowledge, SPARTAN is the first method that learns local causal structures beyond simple state-based settings. Compared to prior transformer world models, our results show that SPARTAN is able to accurately identify causal edges, resulting in significantly improved robustness against distractors, few-shot adaptation, and interpretability.

## 2 Background

We begin this section by introducing our problem setting and providing a brief outline of causal graphical models and causal discovery methods. These are then discussed in relation to our motivation in section 2.3, where we note in particular the role of causality in world models. The goal of our work is to learn a causal model that predicts future states from observation. In order to model interventions, we train our model on a set of intervened environments with *unknown* intervention targets. Specifically, the training data of our model consists of observation sequences sampled from different environments $\{\mathbf{x}^0, \mathbf{x}^1, ..., \mathbf{x}^T, I\}$, where $x^t$ is the observation at timestep $t$ and $I$ is the

environment index. Here, an intervention amounts to altering the dynamics such as changing the friction of an object. We assume access to a representation model that maps observations to factored latent states. $\mathcal{S} = \mathcal{S}_i \times ... \times \mathcal{S}_N$, such that each $\mathcal{S}_i$ is an embedding that represents the state of an entity such as an object in the scene. Examples of such representation models include object-centric models [28, 10] or disentangled causal representations [41, 27, 24] where each $\mathcal{S}_i$ is an object slot or a causal variable respectively. Our aim is to learn a transition function $p(s^{t+1}|s^t)$ that captures the local causal structure.

## 2.1 Causal Graphical Models

A causal graphical model (CGM) [33] consists of a set of random variables $\mathcal{V} = \{V_1, ..., V_N\}$, a directed acyclic graph $\mathcal{G} = (V, E)$ over the set of nodes $\mathcal{V}$ and a conditional distribution for each random variable $V_i$, $P(V_i|Pa(V_i))$, where $Pa(V_i)$ is the set of parents of $V_i$. Each edge in the graph $(i, j)$ means that $V_i$ is a cause of $V_j$. The joint distribution over the variables can be factorised as

$$p(v_1, ..., v_N) = \prod_{i=0}^{N} p(v_i|Pa(V_i)). \tag{1}$$

One important feature of CGMs is that they support *interventions* which are local changes to the distribution that correspond to external changes to the data generation process. Given the intervention $\mathcal{I}$ and its targets $T^{\mathcal{I}} \subset \mathcal{V}$, the joint distribution is

$$p^{\mathcal{I}}(v_1, ..., v_N) = \prod_{v_i \notin T^{\mathcal{I}}} p(v_i|Pa(V_i)) \prod_{v_j \in T^{\mathcal{I}}} p'(v_j|Pa(V_j)), \tag{2}$$

where $P'(\cdot)$ is the intervened conditional distribution. In contrast to a probabilistic graphical model, given a set of interventions, a CGM defines a *family* of distributions.

Recent works in causal discovery, which aim to learn causal graph structure from data, formulate the identification of causal structures as sparsity-regularised optimisation [55, 6]. Intuitively, this amounts to pruning spurious edges and interventions and finding a model that can explain the data with the *sparsest* causal graph. Building on these works, we use sparsity regularisation as an inductive bias to learn *local causal graphs*.

## 2.2 Local Causal Models

In the context of modelling physical systems, since object interactions are often temporally sparse, we further define *local causal models* [34, 42, 15]. Suppose we observe the values of a subset of variables, $\mathcal{V}^{obs} \subset \mathcal{V}$, $\mathcal{V}^{obs} = v^{obs}$. For some neighbourhood $\mathcal{L}$ containing $v^{obs}$, the induced local causal graph $\mathcal{G}^{\mathcal{L}}$ is constructed by removing all edges $(i, j)$ from the global causal graph $\mathcal{G}$ where $V_j \perp\!\!\!\perp V_i|Pa_i(V_j)\backslash V_i$ within the neighbourhood, i.e., the local graph is a subgraph of the global graph obtained by removing any "inactive" edges based on the observed state. Note that the structure of the local graph is now dependent on the observed variables. Concretely, in the context of transition functions, the set of observed variables is the states of the current timestep $s^t$. As a simple example, consider two billiard balls. If it is observed that they are far apart, then there is no local edge, since they cannot influence each other. However, when they are about to collide, their subsequent states will depend on each other, and there is now a *local* edge between them.

Analogously to standard causal discovery, our goal is to infer the sparsest graph at each timestep, such that the next observation can be explained. In the local case, the optimisation objective for *local* causal discovery is to minimise the *expected* number of edges and targets with respect to the data distribution, where the causal edges and intervention targets are state-dependent and are built dynamically at each timestep. [15] provides a theoretical analysis of the conditions under which local causal structures can be identified using sparse regularisation.

## 2.3 Causality and World Models

Conceptually, we argue that world models should be structured to reflect the underlying sparse and local causal structure of the dynamics to perform efficient adaptation. To illustrate our point, consider the setting of traffic behaviours: changes in traffic rules arise when travelling to countries where

vehicles drive on the opposite side of the road. Although the relative position of vehicles given the road boundaries may change, most traffic behaviours (e.g., stopping at a red light or lane keeping) would remain the same. A world model that can efficiently adapt should update only a small subset of learnt dynamics (i.e., the relative position of vehicles to the road boundary) while leaving other dynamics unchanged. Our proposition is motivated by the sparse mechanism shift hypothesis [31, 41], which states that naturally occurring distribution shifts can be attributed to sparse interventions on causal mechanisms. Formally, this means that reasonable changes in the environment can be modelled by changing a small subset of the conditional distributions in Eq. 1 while the rest remain invariant. In the special case where interventions act only on variables that are not causally related to model predictions, e.g., removing irrelevant objects in a scene [39], models that reflect the correct causal structure would remain robust.

While learning a global causal graph is sufficient in some settings, in the context of dynamics models, such a graph is often close to fully connected since any entities that interact with each other at *any point* are connected in the graph regardless of how unlikely the interaction is. Returning to the traffic example, a global causal graph would connect every vehicle in the scene, as they can all influence each other when close together. However, events such as "vehicle A causes vehicle B to stop" can be more appropriately captured by *local causal graphs* with the edge $A \rightarrow B$ but not by edges to other vehicles that are irrelevant at the time. As such, we argue that local causal graphs are more suitable for models to fully exploit the sparse structure of the problem at hand. In the following section, we present SPARTAN, a transformer-based instantiation of sparsity-regularised local causal discovery.

## 3 Sparse Transformer World Models

Our goal is to develop a world model that learns *local* causal models as transition functions. We build on the Transformer architecture [48] as it achieves state-of-the-art prediction performance in object-factored world models [53]. Moreover, its attention mechanism provides a natural way to control the flow of information between object tokens. While prior work [34] argues that soft attention is a strong enough inductive bias for learning local causal graphs and proposes a thresholding heuristic on the attention patterns, we posit that this is insufficient beyond simple state-based scenarios and that appropriate sparsity regularisation and hard attention play a crucial role in scaling this up to more complex, observation-based environments. To demonstrate with a simple example, consider a system with only one causal edge, $s_i^t \rightarrow s_j^{t+1}$ and many other irrelevant nodes. A soft-attention model can predict $s_j^{t+1}$ as long as the attention value for $s_i^t$ is non-zero, i.e., information can flow from the $i$-th token to the $j$-th token. However, the model is free to have non-zero attention values over other tokens. In this case, there is no guarantee that the attention value on $s_i^t$ is the highest amongst all tokens, and therefore, applying a threshold on the attention values may 'catch' spurious edges or miss necessary edges. Instead, we need a model that *masks* the information flow between the tokens and penalises connections between tokens. Any model that contains spurious connections would have a high loss due to the sparsity penalty, while models that do not contain the $s_i^t \rightarrow s_j^{t+1}$ would suffer from low prediction accuracy. Under this scheme, the model that achieves the best training loss would be the one that contains only the correct causal edge. In the following section, we present *SPARrse TrANsformer World models* (SPARTAN), an instantiation of a masked transition model using a Transformer-based architecture with *hard* attention.

### 3.1 Learnable Sparse Connections

We start with the architecture of a single layer before extending the apporach to the multi-layer case. Given object representations $s_{1:N}^t$ as input, we obtain the keys, queries, and values, $\{k_i, q_i, v_i\}$ via linear projection, as in the standard Transformer. Using these, we sample an adjacency matrix,

$$A_{ij} \sim \text{Bern}(\sigma(q_i^T k_j)), \tag{3}$$

where $\text{Bern}(\cdot)$ is the Bernoulli distribution and $\sigma$ is the sigmoid function. The hidden features are then computed using the standard scaled dot-product attention with the adjacency matrix as masks.

$$h_i = \sum_j \frac{A_{ij} exp(q_i^T k_j / \sqrt{d_k}) v_j}{\sum_i exp(q_i^T k_j / \sqrt{d_k})}, \quad \hat{s}_i^{t+1} = MLP(h_i + s_i), \tag{4}$$

where $\hat{s}_i^{t+1}$ is the prediction for the $i$-th object in the next timestep. Here, the adjacency matrix acts as a mask that disallows information flows from tokens that are not in the parent set of the querying

token. Note that the adjacency matrix is dependent on the current state via the key-query pairs. The learnt adjacency matrix can therefore be interpreted as the local causal graph, i.e. $A_{ij} = 1$ means $s_j^t \in Pa^{\mathcal{L}}(s_i^{t+1})$. The sampling step is made differentiable via Gumbel softmax trick [17]. Typically, multiple Transformer layers are stacked sequentially for expressiveness. This presents a problem for the learnt adjacency matrix, as information can flow between tokens in a multi-hop manner across different layers: for example, token $i$ can influence token $j$ via $i \to k \to j$ without the edge $(i, j)$ being present in any of the adjacency matrices. SPARTAN mitigates this problem by tracking the number of *paths* from token $i$ to $j$. Concretely, for $L$ layers, we compute the path matrix,

$$\bar{A} = (A^{L+1} + \mathbb{I})...(A^2 + \mathbb{I})(A^1 + \mathbb{I}), \tag{5}$$

where $A^l$ is the adjacency matrix at layer $l$ and $\mathbb{I}$ is the identity matrix. The identity matrix is added due to residual connections. The path matrix $\bar{A}$ has the property that $\bar{A}_{ij}$ is the number of paths that lead from $j$ to $i$. In this case, $s_j^t$ is a local causal parent of $s_i^{t+1}$ iff $\bar{A}_{ij} >= 1$.

## 3.2 Interventions

We can extend our model to represent interventions in cases where the model is trained on a dataset with a set of environments with intervened dynamics.[1] During training, the model has access to the environment index $I$ for each transition but does not know which object is affected by the intervention. Conditioned on the current state $s_{1:N}^t$ and the environment index $I$, the model needs to identify a subset of $s_{1:N}^{t+1}$ as intervention targets and change the predictions (cf. Eq. 2). To do this, we keep a learnable codebook of *intervention tokens*, $\mathcal{T}_{1:K} \in \mathbb{R}^{d \times k}$, where $d$ is the dimension of the state tokens and $K$ is the number of interventions. We append the corresponding intervention token $\mathcal{T}_I$ to the object tokens and proceed as described in the previous subsection. Here, $\bar{A}_{i(N+1)} = 0$ indicates that there is no path from the intervention token to $s_i^{t+1}$ and therefore $s_i^{t+1}$ is not an intervention target, i.e., the prediction for $s_i^{t+1}$ is not affected by the changes in this environment.

At test time, the model adapts to unknown interventions by observing sequences without the environment index. We perform adaptation by finding an intervention token $T_{adapt} \in \mathbb{R}^d$ that best fits the observed data via gradient descent in the token space. Note that the adaptation intervention token need not be in the discrete set of intervention tokens from training and can therefore model unseen environments. In Sec. 4, we show that this approach can generalise to the composition of previously seen interventions and leads to efficient adaptation.

## 3.3 Training

Our aim is to fit the data distribution with the sparsest possible model in terms of causal edges and interventions. The training objective is to minimise the expected loss with sparsity regularisation,

$$\min_{\theta} \mathbb{E}_{s^t, s^{t+1}, I}\left[\mathcal{L}(\hat{s}^{t+1}) + \lambda_s |\bar{A}|\right], \tag{6}$$

where $\theta$ are the model parameters, including the parameters for the Transformer and the intervention tokens, $\mathcal{L}$ is a loss function such as MSE, and $\lambda_s$ denotes the regularisation hyperparameter. Note that due to the intervention token being part of the input during training, $|\bar{A}|$ is the sum of the number of causal edges *and* the intervention targets. In practice, the model is sensitive to the choice of $\lambda$ since a high $\lambda$ can lead to mode collapse. We alleviate this via Lagrangian relaxation, which schedules the regularisation weight. The details of this setup and other training information are in App. A.

## 4 Experiments

Our experiments are designed to investigate the following guiding questions:

1. Does sparsity enable local causal discovery from observations?
2. Can the sparse model match the prediction accuracy of fully-connected models?
3. Does learning interventions lead to improved robustness and adaptation sample efficiency?

---

[1]Note that this is an extension, rather than a requirement, to our approach to capture multi-environment training. In single environment cases, SPARTAN can be directly applied as described above.

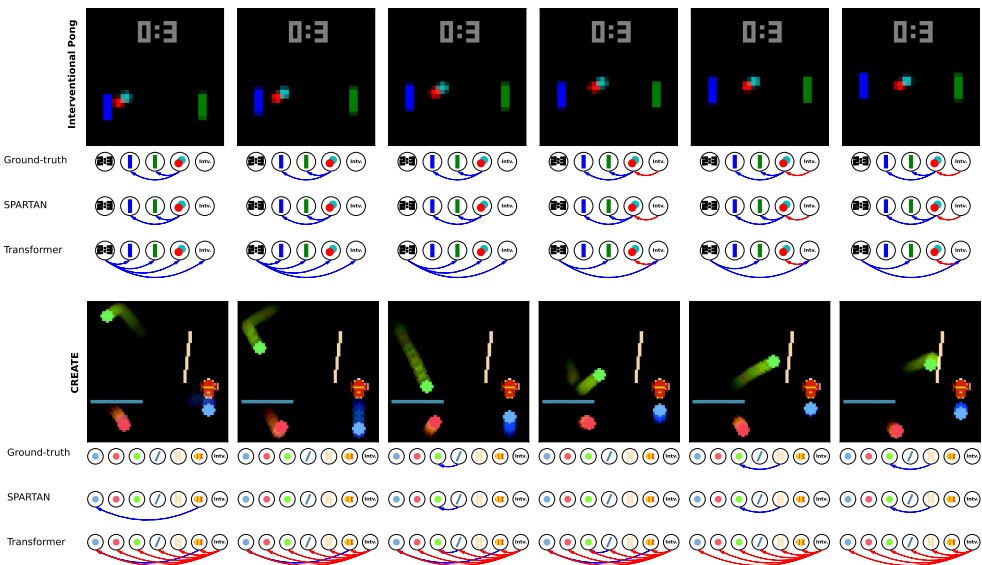

Figure 2: Example rollouts in the two simulated environments with the learnt local causal graph visualised. Blue and red arrows between icons indicate the learnt causal edges and intervention targets respectively. In the *Interventional Pong* example, the intervention is that the ball slows down in the middle. SPARTAN correctly identifies the same causal dependencies as the ground-truth (e.g. ball causes the paddles to follow). The Transformer baseline learns edges that do not correspond to the ground-truth. Similarly in *CREATE*, SPARTAN learns the correct causal edges (e.g. green ball bounces off the blue plank) while Transformer learns many spurious interventions.

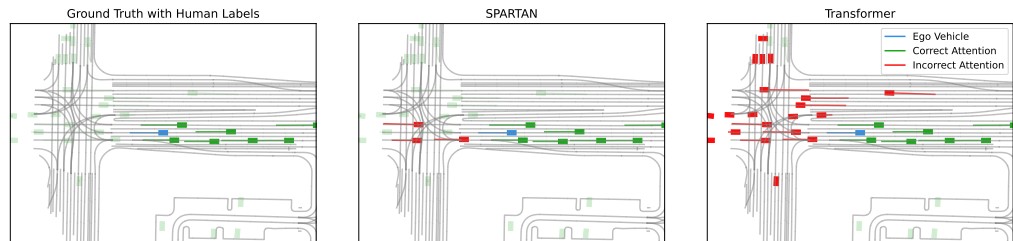

Figure 3: Visualisation of the causal relationships learned by the models compared to human labeled data. The ego vehicle is blue. Transparent rectangles means that the vehicle is not a causal parent of the ego vehicle. SPARTAN learns a similar attention pattern to human data (e.g. focus on vehicles that are moving in the same direction) whereas Transformer learns many spurious edges.

**Datasets** We evaluate our model in three domains: *Interventional Pong*, *CREATE*, and *Traffic*. The *Interventional Pong* dataset [27], a standard benchmark for causal representation learning, is based on the Pong game with interventions. The *CREATE* [16] environment is a 2D physics simulation that consists of interacting objects such as ladders, cannons, and balls. This is similar to the PHYRE dataset [2] on which SlotFormer [53] is evaluated, but it has more interaction types, allowing for more interventions to better showcase the capabilities of SPARTAN. For these two domains, we obtain object slot representations by providing ground-truth masks for objects and encoding each object separately using a VAE [21]. In order to evaluate our model on more realistic tasks, the *Traffic* domain uses the Waymo Open Dataset [44] which is collected in real life. The task is to predict the motion of the ego vehicle given the observed vehicle trajectories (i.e., past positions) and map layout lines. For evaluating the accuracy of the learnt causal graphs, we compare them against ground-truth causal graphs for the two simulated domains. For the *Traffic* domain, we compare against human-labelled causal graphs [39]. In the simulated domains, we also train the models using interventional data, as described in Sec. 3.2. Interventions are defined as changes to the simulated dynamics such as changing the strength of gravity. See App. B for details.

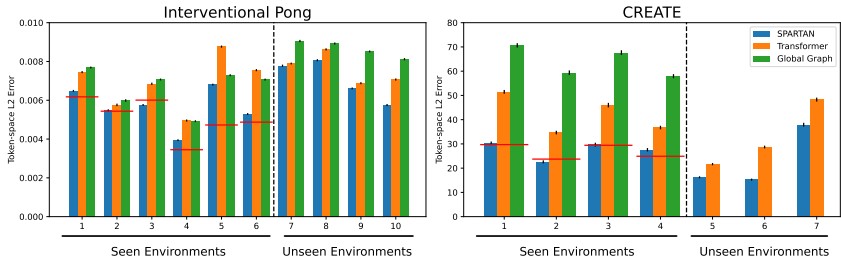

Figure 4: Adaptation errors on the two datasets. Each model has access to 5 trajectories with unknown environment index. "Unseen environments" refers to interventions that are not in the training set. Red lines indicates the prediction error if the environment index is provided, which serves as a lower bound. SPARTAN (blue) consistently achieves the lowest errors across environments.

**Baselines** In order to investigate whether SPARTAN can maintain the state-of-the-art prediction accuracy achieved by Transformer-based models, we compare our method against a *Transformer* baseline using a Transformer for learning dynamics. This can be seen as an instantiation of the SlotFormer [53] architecture with a one-step context length, except that the slots are learnt using ground-truth object masks. For the *Traffic* domain, we use MTR [43], a state-of-the-art transformer-based model designed for traffic data as the base architecture. In terms of causal discovery, existing methods [51, 15] only operate on state observations where each state is represented as a scalar and are therefore not applicable in our setup, which observes object embeddings. We provide a more detailed discussion on the applicability of these prior approaches and offer extra experimental comparisons in App. C. Closest to our work is CoDA [34], which argues that soft attention in Transformers alone is sufficient for learning local graphs and proposes a thresholding heuristic. We compare against this approach and show that sparsity regularisation in SPARTAN is crucial in local causal discovery. To corroborate that local causal graphs are more suitable for modelling physical interactions, we further compare against the *Global Graph* baseline, which is based on the transition function proposed in VCD [25] and AdaRL [14], which learns a fixed global causal graph using masked MLPs. The details of the baseline models are provided in App. A.

Table 1: Rollout prediction error and average SHD between the learned graphs and the ground-truth causal graphs. Lower is better.

| **Method** | Intv. Pong | | CREATE | | Traffic | |
|---|---|---|---|---|---|---|
| | Pred. Err. | SHD | Pred. Err. | SHD | Pred. Err. | SHD |
| SPARTAN (Ours) | **8.60** | **1.51** | **246.6** | **1.17** | 0.52 | **6.84** |
| Transformer | 8.83 | 6.37 | 265.5 | 6.29 | **0.51** | 10.6 |
| Global Graph | 11.36 | 5.42 | 277.4 | 8.77 | - | - |

## 4.1 Graph Learning

The prediction accuracy of the models is shown in table 1. Overall, SPARTAN is able to make predictions at a similar level of accuracy to and often outperforms the Transformer baseline. We investigate whether casual discovery via sparsity regularisation is required to identify correct local causal graphs in the setting of dynamics modelling. While [34] suggests that extracting causal graphs by thresholding attention patterns is sufficient in simple environments, here we demonstrate that in more complex environments, learning *hard* attention patterns via sparsity regularisation is *necessary* for accurate local causal discovery. Fig. 2 visualises example rollouts with the local causal graphs.[2] Here, we observe that SPARTAN can reliably recover the relevant causal edges and intervention targets at each timestep. In the Interventional Pong environment, our method identifies that the paddles are constantly influenced by the ball as they follow it. In contrast, the Transformer baseline

---

[2]When evaluating the Transformer baseline, we pick a threshold that achieves the best structural hamming distance. Note that this is not possible in practice without access to ground-truth information.

gives erroneous edges such as "Score → Ball" (frame 1). Similarly, in the CREATE environment, SPARTAN accurately discovers sparse interactions between objects. In the *Traffic* domain, Fig. 3 shows that SPARTAN learns to attend to vehicles in neighbouring lanes, which is largely consistent with the human labels, whereas the Transformer baseline attends to many irrelevant vehicles. In this case, heuristics based on spatial proximity would also fail since there are vehicles that are far ahead that are relevant while vehicles that are close by but in the opposite lane, for example, should be ignored. This further highlights the importance of learning local causal graphs. Quantitatively, we evaluate the Structural Hamming Distance, a commonly used metric in graph structure learning, between the learned graphs and the ground-truth. Table 1 shows that SPARTAN achieves significantly lower distance compared to the baselines across all domains. We do not apply the Global Graph baseline to the *Traffic* domain since the fixed graph approach cannot be extended to include a different number of objects.

## 4.2 Robustness and Adaptation

Table 2: Percentage change of error when non-causal entities are removed. Lower is better.

|  | SPARTAN | Transformer | Local Attn. | Global Graph |
|---|---|---|---|---|
| Intervention Pong | $24.5 \pm 4.4$ | $1140.2 \pm 15$ | - | $\mathbf{22.2} \pm 1.3$ |
| CREATE | $\mathbf{28.6} \pm 1.7$ | $138.2 \pm 5.6$ | - | $73.6 \pm 1.3$ |
| Traffic | $\mathbf{13.9} \pm 0.2$ | $32.8 \pm 0.4$ | $25.5 \pm 0.3$ | - |

[39] proposed evaluating the robustness of models by considering the change in the prediction error when non-causal entities are removed from the scene. This can be seen as a special case of interventions that target only variables that are not the causes of the predicted variable. Concretely, for each object in the scene, we remove any object that is not in the ground-truth parent set and compute the mean absolute value of the percentage change in the prediction error. Here, a model that correctly captures the causal structure would be robust, whereas a model learning spurious correlations would have large fluctuations. Table 2 shows that, compared to a non-regularised Transformer, SPARTAN is significantly more robust, suggesting that the Transformer overfits to spurious correlations between entities. In the *Interventional Pong* domain, the ground-truth causal graph is largely consistent across time steps. As such, the Global Graph baseline is sufficient in capturing the causal structure and hence remains robust. However, this degrades in the more complex CREATE domain, where the local causal graph is more state-dependent, while SPARTAN achieves the best robustness. In the Traffic domain, we also implement a local attention variant of the Transformer baseline, proposed in MTR [43], which applies a spatial K-nearest neighbour mask on the attention pattern between the tokens. Whilst this heuristic improves the model, SPARTAN remains significantly more robust to non-causal changes. A wide range of model architectures is evaluated in [39], with reported changes ranging from 25% to 38%, which is consistent with our findings. Remarkably, the authors in [39] propose a data augmentation scheme using ground-truth information, resulting in a change of 22%, which is still significantly worse than what SPARTAN achieves, highlighting the efficacy of our method.

To investigate adaptation efficiency, we train the models to adapt to a sample of five trajectories from an intervened environment. For SPARTAN, we perform adaptation as described in Sec. 3.2. We follow a similar procedure of optimising over intervention tokens for the Global Graph baseline. For the Transformer baseline, adaptation is performed via gradient finetuning on the provided trajectories. We also adapt the models to previously *unseen* environments where the interventions performed are not included in the training set. In the Interventional Pong dataset, the new environments are obtained by combining interventions from the training set, e.g. intervening on the ball and the motion of the paddle at the same time. In CREATE, we change the scene composition as well as the dynamics, i.e. we change the number of objects in the scene. In this scenario, the Global Graph baseline cannot be applied. We present the results in Fig. 4, which shows that SPARTAN consistently outperforms the baselines in this few-shot adaptation setting.

# 5 Related Works

Our work is situated within the context of learning world models [11, 12]. In particular, we build on methods that learn dynamics models over object-centric representations [10, 28, 52]. In this space, various architectures have been proposed to model the interactions between object slots as message passing, including GNNs [22, 45], RNN [47, 18] and pair-wise interactions [49]. SlotFormer [53] achieves state-of-the-art results by applying a Transformer-based dynamics model on pre-trained object-slots [28]. We build on this approach by extending the Transformer architecture with learnable sparse masks, resulting in improved robustness, adaptability, and interpretability without compromising prediction performance or the flexibility to model varying numbers of objects.

Our approach is motivated by recent advances in causal machine learning [40, 19]. In particular, [37] provides a theoretical account of how identifying a causal world model leads to generalisable agents. Concretely, our sparsity regularisation-based approach is motivated by optimisation-based causal discovery methods [6] that use sparsity as an inductive bias for learning causal relationships. In this vein, VCD [25] and AdaRL [14] learn fixed causal graphs using sparsity regularisation in the context of world models. We extend these approaches by learning state-dependent *local* graphs that are better suited for modelling physical interactions. The notion of *local causal model* for dynamics is discussed in [35, 34, 51], which demonstrates the potential benefits of local causal models in terms of data-efficiency, robustness, and exploration. The CoDA family of works [34, 35, 1] uses local causal graphs to enable the generation of counterfactual data for data augmentation. However, estimating a local causal model from data remains challenging. Several approaches to learning local causal structures have been proposed. CAI [42] uses conditional mutual information to estimate the local influence that an agent has on the environment. ELDEN [51] uses sparsity-regularised partial derivatives between states based on a learnt dynamics model to infer local causal graphs. FCDL [15] uses sparsity regularisation to learn a quantised codebook of possible causal connections. The key difference between these approaches and ours is that they require state observations, whereas our method can operate on object-centric image embeddings. Moreover, owing to the flexibility of transformers, SPARTAN is able to learn in more realistic scenarios where the number of objects or the scene composition changes between data samples. We provide a more detailed discussion on the applicability of existing methods in our setting and present additional experiments in App. C. Closest to our work is [34] which uses attention patterns in transformer layers as causal edges. We empirically show that our model achieves significantly improved accuracy in terms of the discovered graph structure.

In a broader context, enforcing sparse connections between tokens in Transformers has also been explored in other settings, such as NLP [9, 8, 26] and computer vision [56]. These methods require pre-defined masks based on domain knowledge, such as paragraph structures. SPARTAN differs from these approaches as it does not require any pre-defined masks and can be seen as a way to learn these masks from data. To this end, investigating the application of sparsity-regularised hard attention in these domains outside of world modelling offers a tantalising avenue for future research.

On a conceptual level, our work is also related to the notion of the Consciousness Prior [3], which advocates for an attention-like mechanism that selects a sparse bottleneck of 'active' entities in any given scene. This idea is realised in [54], showing that the sparsity induced by the Consciousness Prior leads to improved generalisation in model-based planning tasks. Our approach differs from this idea in two principal ways: 1) [54] requires pre-specifying the number of active entities, whereas the sparse attention in our approach flexibly attends to a varying number of objects in a state-dependent manner. 2) In the Consciousness Prior framework, the model is sparse in the sense that it chooses a small subset of objects that is active. In contrast, we consider a complementary kind of sparsity, where we model the dynamics of all objects but use sparse attention to determine how each object depends on other objects.

# 6 Conclusion

We tackle the problem of adaptation in world models through the lens of local causal models. To this end, we propose SPARTAN, a structured world model that jointly performs dynamics model learning and causal discovery. We show on image-based datasets that attention mechanisms alone are not sufficient for discovering causal relationships and develop a novel sparsity regularisation scheme

that learns accurate causal graphs, resulting in significantly improved interpretability and few-shot adaptation capabilities without compromising prediction accuracy.

**Limitations and future work.** There are several limitations to our approach that serve as pointers for future investigations. 1) We empirically show that local causal graphs can be learnt from data. While the use of sparsity regularisation is grounded in prior works that have theoretical guarantees [15], we do not make theoretical statements about the identifiability of local causal graphs in the setting where the scene composition can vary between each data sample. Future work should explore the conditions under which local graphs can be identified. 2) During adaptation, SPARTAN adapts by optimising over the learnt 'intervention space'. We have shown that this is sufficient to generalise to combinations of seen interventions or different numbers of objects. However, in more extreme cases where the test time environment contains completely new behaviours (e.g. ball teleporting), there might not be a corresponding intervention token in the space, while a finetuning approach would be able to converge to the right behaviour (given enough data). One avenue of exploration is to consider procedurally generating intervened environments, akin to domain randomisation [32, 46], to cover the space of all meaningful interventions. 3) Our approach relies on pre-disentangled object representations. Note that while we have used ground-truth segmentation masks for objects in our evaluations, pre-trained object-centric representations can also be utilised. We provide further discussion on this in App. D. An interesting extension of our work is to investigate whether local sparsity can *induce* the emergence of disentangled causal representations [25, 23] by jointly training an encoder with the dynamics model.

### Acknowledgments

This research was supported by an EPSRC Programme Grant (EP/V000748/1). We would also like to thank the University of Oxford Advanced Research Computing (ARC) (http://dx.doi.org/10.5281/zenodo.22558) and the SCAN facility in carrying out this work.

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

# A Model Details

## A.1 Hyperparameters

SPARTAN and the Transformer baseline are implemented as stacked transformer encoder layers with residual connections. The Global Graph baseline is implemented as an ensemble of MLPs with learnable adjacency matrix masks. This is similar [25, 14] to the modification that our baseline works on object embeddings rather than scalar states. Each MLP predicts a separate object embedding. For each MLP, the input object tokens are masked according to a learnable adjacency matrix before being passed through the model. A sparse regularisation is applied to the adjacency matrix. The hyperparameters for SPARTAN and the baselines are shown in table 3 and 4. For the experiments on the simulated datasets, all models are trained on single GPUs (mixture of Nvidia V100 and RTX 6000) and converge within 3 days.

For the Traffic domain, predictions need to be multimodal since each vehicle has multiple possible trajectories. This is common for the dataset we consider. To overcome this, we use MTR [43] as the base architecture. MTR has two stages: first, it uses self-attention between tokenised map lines and vehicle trajectories; second, it applies cross attention using multiple queries, each representing one possible mode of motion. The output for each query is then used as a Gaussian mixture model to predict multimodal motion patterns. We adapt SPARTAN to this architecture by swapping all attention layers with the sparsified version proposed in this paper. The original work also proposes a local attention masking scheme for each layer, which only allows each token to attend to the k-nearest neighbour based on the position of each object. We refer to this as Local Attn in table 2. In the traffic domain, the models are trained in parallel on 4 GPUs due to the size of each scene (roughly 1000 tokens). Training takes less than one week for the baseline MTR model and under two weeks for SPARTAN.

Table 3: Hyperparameters for SPARTAN and the Transformer Baseline.

| Hyperparameter | Interventional Pong | CREATE |
|---|---|---|
| Token Dimension | 32 | 64 |
| Embedding Dimension | 512 | 512 |
| n. transformer layers | 3 | 3 |
| MLP hidden dimension | 512 | 1024 |
| n. MLP layers per transformer layer | 3 | 3 |
| lr | 5e-5 | 5e-5 |
| Optimiser | Adam[20] | Adam |

## A.2 Lagrangian Relaxation

As discussed in Sec. 3.3, the model is sensitive to the choice of $\lambda$. Since removing non-causal edges should not deteriorate the prediction accuracy, we formulate a constrained optimisation problem where we minimise $|\bar{A}|$ under the constraint that $\mathcal{L} \leq \tau$, where $\tau$ is the target loss,

$$\min_{\theta} \quad \mathbb{E}\big[|\bar{A}|\big] \quad s.t. \quad \mathbb{E}\big[MSE(s^{t+1}, \hat{s}^{t+1})\big] \leq \tau. \tag{7}$$

Table 4: Hyperparameters for the Global Graph Baseline.

| Hyperparameter | Interventional Pong | CREATE |
|---|---|---|
| Token Dimension | 32 | 64 |
| MLP hidden dimension | 1024 | 1024 |
| n. MLP layers | 5 | 5 |
| lr | 5e-5 | 5e-5 |
| Optimiser | Adam | Adam |

We set the target as the loss achieved by a fully connected model. Applying Lagrangian relaxation [5], we obtain the following min-max objective,

$$\max_{\lambda > 0} \min_{\theta} \quad \mathbb{E}\big[|\bar{A}|\big] + \lambda\bigg(\mathbb{E}\big[MSE(s^{t+1}, \hat{s}^{t+1})\big] - \tau\bigg). \tag{8}$$

We solve this by taking gradient steps on $\theta$ and $\lambda$ in an alternating manner. To ensure $\lambda$ is positive, we perform updates in the form of $\lambda \leftarrow \alpha * exp(MSE - \tau) * \lambda$, where $\alpha$ is the step size. Intuitively, this has the effect of increasing $\lambda$ when the error is higher than the target, i.e. weighting the error term when the error is high. This is analogous to the GECO [36] framework for tuning the KL loss in a VAE. In practice, we initialise $\lambda$ to be high and set $\tau$ to be the error achieved by the fully connected model. We use a moving-average estimator for the $MSE - \tau$ term for stability. We also rewrite the training objective to $(MSE - \tau) + |\bar{A}|/\lambda$ so that the loss value is more stable. This acts as a scheduling scheme where the optimisation focusses on learning the dynamics and switches to pruning redundant edges once the error is below the target. Fig. 5 shows example training curves demonstrating the training dynamics of SPARTAN. At the start of the training, $log(\lambda)$ remains low as the model prediction error improves. This allows an increase in the number of active edges. As the prediction error becomes low enough, the sparsity penalty automatically increases, and the number of edges gradually decreases while maintaining prediction accuracy.

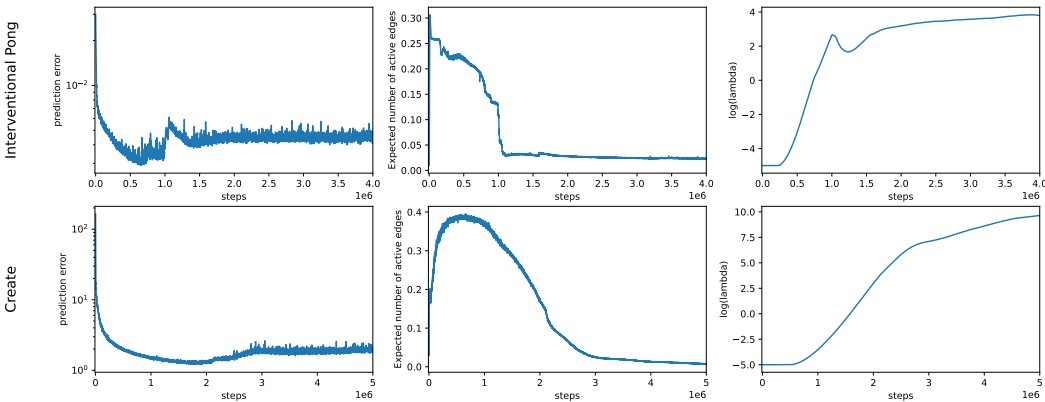

Figure 5: Example training curves.

# B  Datasets

## B.1  Interventional Pong

The *Interventional Pong* environment is based on the Pong game and was originally developed in [27] for investigating causal representation learning. In the original work, the set of interventions

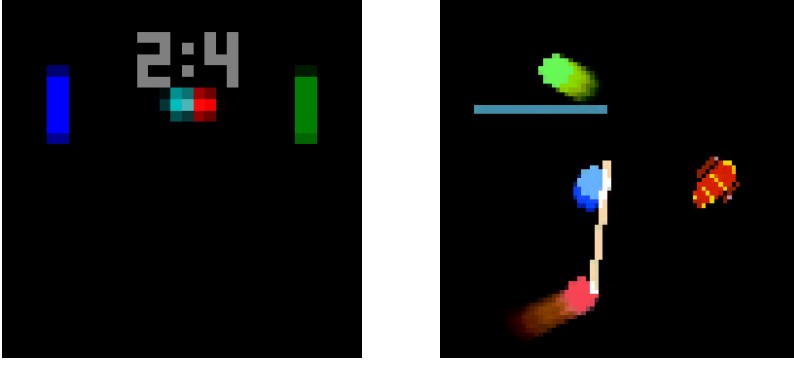

Figure 6: Example observations in the two environments.

Table 5: Environment configurations for Intervention Pong.

|       | Env ID | Description |
|-------|--------|-------------|
|       | 0      | No intervention |
|       | 1      | Ball follows curved trajectories |
|       | 2      | Left paddle moves away from ball instead of towards |
| Seen  | 3      | Right paddle mirrored along y-axis |
|       | 4      | Strong friction on the ball in the middle section of the field |
|       | 5      | Changed bouncing dynamics when paddles hit the ball |
|       | 6      | Turn on gravity on the left side of the field |
|       | 7      | Gravity on the left side + changed bouncing dynamics with paddles |
| Unseen| 8      | Strong Friction in the middle + changed bouncing on left side only |
|       | 9      | Curved ball trajectory + left paddle moves away from ball |
|       | 10     | Gravity on the left side + right paddle moves away from ball |

Table 6: Environment configurations for CREATE

|        | Env ID | Objects | Description |
|--------|--------|---------|-------------|
|        | 0      |         | No intervention |
|        | 1      |         | Disabled gravity |
| Seen   | 2      | {Red, Blue, Green} balls, wall, ladder cannon | Balls go *down* ladders instead of up |
|        | 3      |         | Increased elasticity of walls |
|        | 4      |         | Disabled cannon |
|        | 5      | {Red, Blue, Green} balls, Wall, 3*Ladders | No interventions |
| Unseen | 6      | {Red, Blue, Green} balls, wall, 3*Ladders | Balls go *down* ladders instead of up |
|        | 7      | {Red, Blue, Green}, 2*Walls, 2*Cannons | Increased elasticity of walls |

included randomly perturbing the positions of the balls and the paddles. We modify the interventions to model dynamics changes, such as adding gravity or changing the way the ball bounces off the paddles, as these are more appropriate in the context of world model learning. A similar setup is also used in [14] in the context of few-shot adaptation. The original Interventional Pong dataset is under the BSD 3-Clause Clear License.

The environment contains 4 objects: the left paddle, the right paddle, the ball, and the score. Here, local causal edges include ball $\rightarrow$ left, right paddles as the paddles follow the ball; paddles $\rightarrow$ ball when the ball collides with the paddles; ball $\rightarrow$ score when the ball crosses the left or right boundary lines. Observations are provided as $32 \times 32 \times 3$ RGB images with masks for each object. To capture velocity information for the ball, the ball image shows the position of the ball over 2 consecutive timesteps in different colours. Each object is separately encoded into 32-dimensional latent object-slots using a VAE [21]. The configurations of different intervened environments are shown in table 5.

## B.2 CREATE

The *CREATE* environment is built using the CREATE simulator [16] which consists of physical interactions between various objects. The simulator is under the MIT License. In our experiments, the objects in the environment include {*blue*, *red*, *green*} balls, a wall, a ladder, and a cannon. The balls collide elastically. All objects are initialised at random positions with random orientations. The wall remains static for each sequence. The ladder object allows the balls to climb up. The cannon object projects the balls at a fixed velocity in the direction it points at. Here, local edges capture when the objects collide or interact with each other. Observations are given as $84 \times 84 \times 3$ images with masks for each object. To capture the velocities of the balls, a trail of the past positions (up to two timesteps) is shown in the image for each ball. The intervened environments are shown in table

6. In the unseen environments, we test the models using different scene composition, i.e. different numbers of objects.

### B.3 Traffic

We use the Waymo Open Dataset [44] for our Traffic domain. The dataset consists of real world scenes, which include map lines (such as lane markings) and vehicle trajectories. The map lines are provided as polylines (i.e. each line is represented as a set of positions). The history of the state of each vehicle is provided as a list of positions, headings and velocities. Up to one second of state is given for each vehicle, and the task is to predict 9 seconds into the future for the centre vehicle. The number of map lines and vehicle trajectories is different for each scene, with roughly 800 map lines and 50 vehicles per scene. This presents a major challenge for prior approaches in causal world model learning. Since the dataset is not generated by simulation, we do not apply interventions to the data.

## C   Relation to existing approaches and extra baselines

The aim of this work is to integrate the inductive bias of sparse interactions, common in causal discovery, into a transformer-based architecture. As such, we focus our evaluations on environments that are commensurate in complexity and scale to the settings on which prior transformer world models are evaluated. While there exist prior approaches for local causal learning, as discussed in the related works section, these methods cannot be readily scaled to the environments in the experiments presented in this paper. This is due to two principal reasons: first, prior methods operate directly on state-based observations, such as x, y positions of objects, whereas in our environments, objects are represented as learnt embeddings (typically 64 dimensions) that are mapped from images; second, and more fundamentally, our environments, in particular CREATE and traffic domains, require the world model to operate on different numbers of objects and different compositions of objects across samples, e.g. different numbers of vehicles across scenes. A transformer-based world model can accommodate these requirements by treating each object embedding as a separate token. Here, we discuss the particular architectural choices of prior works that hinder their application in these environments.

**ELDEN** [51] explores the role of local causal models in the context of intrinsic motivations for RL. Here, the forward dynamics model is trained with a sparsity regularisaion, proxied by using the L1 norm, on the partial derivatives between the states. In principle, this technique can be extended beyond scalar states to the case in which each object is represented by higher dimensional embedding vectors. However, we implemented ELDEN for our experiments and found that, since the loss is a function of many partial derivatives, the optimisation process becomes prohibitively memory-intensive and highly unstable.

**FCDL** [15] investigates the robustness improvements afforded by local causal models. This approach learns a *codebook* of possible local causal graphs and infers the correct structure conditioned on the state. Sparse regularisation is then applied to the set of local causal graphs represented by the codebook. The drawback of this approach is that the local graphs must have the same set of nodes across samples, meaning that all scenes must have exactly the same composition of objects.

**CAI** [42] considers a related but different setting from our method, where the goal is to estimate the causal influence of an agent on the states of the objects in the scene. This approach relies on testing the conditional independence using conditional mutual information estimation between the agent's action and the objects based on a learned dynamics model. However, estimating conditional mutual information is challenging in the multivariate case.

**CODA** [34] uses threshold heuristics to determine local structures. We compare against this in our main experiments and show that attention alone is not able to reliably learn the correct dependency structure.

Beyond the local causal structure learning literature, the idea of learning *context-dependent* causal structures is also explored in *Amortised Causal Discovery* (ACD) [29] which uses a GNN backbone to infer causal structures conditioned on the history of states in a time-series setting. This approach can be adapted to the *state-dependent* local causal structure case by conditioning the graph encoder on the current state rather than on an entire time-series. In table 7, we compare the graph learning

Table 7: Average SHD between the learned graphs and the ground-truth. Lower is better.

|  | SPARTAN | ACD | Sparse ACD |
|---|---|---|---|
| Intervention Pong | **1.50** | 10.27 | 2.67 |
| Create | **1.17** | 39.77 | 5.45 |

accuracy between SPARTAN and ACD. We note that ACD requires setting a prior level of expected sparsity. Here, we compare against a naive choice of 50% sparsity (labelled ACD) as well as the sparse baseline, which uses the ground-truth sparsity as a prior (labelled Sparse ACD). The results show that SPARTAN significantly outperforms this baseline, even if the true sparsity level, which is in practice not accessible, is specified. This further corroborates the efficacy of our model.

# D Using pretrained object-centric representations

Many existing works that aim to capture object interactions, including SlotFormer [53], use pre-trained object-centric representations [28] that handle the segmentation and tracking problems. Our approach can be readily applied to any such pretrained representations. In our experiments, we opted to use ground-truth masked object representations for the principal reason that they enable the evaluation of the learned graphs: quantitatively evaluating the SHD of learned graphs requires mapping each latent embedding to the corresponding ground-truth object. Learned object embeddings can make this mapping noisy and ambiguous due to imperfect segmentations. Since learning high-quality representations is not the main focus of this work, we use ground-truth masked object representations to ensure the fairness and validity of our evaluation. To demonstrate the generality of our results, we have trained our model using a learned slot representations in the pong environment. Here, to measure the SHD between the learned graph and the ground-truth causal graph, we use the visualised attention masks from Slot Attention to map the latent object representations to ground-truth objects. In table 8, we present the results, which are consistent with the main findings in the paper: our model can achieve the same level of prediction while capturing the underlying graph structure more accurately.

Table 8: The prediction error and SHD of the learned graphs in the Intervention Pong environment.

|  | SPARTAN | Transformer |
|---|---|---|
| Prediction Error ($\pm$ SE) | $7.21 \pm 0.69$ | $6.90 \pm 0.83$ |
| SHD ($\pm$ SE) | $\mathbf{2.69} \pm 0.02$ | $8.52 \pm 0.02$ |

