# OpenReview forum: "SPARTAN: A Sparse Transformer World Model Attending to What Matters"
_NeurIPS.cc/2025/Conference — NeurIPS 2025 poster_

### Official Review · Reviewer_gSAB · 2025-06-15

**Clarity:** 3
**Significance:** 3
**Originality:** 3
**Rating:** 4
**Confidence:** 4

**Summary:**

This paper introduces SPARTAN, a Transformer-based world model designed to learn the underlying dynamics of multi-object systems from observations. The central argument is that world models should capture the sparse and local nature of physical interactions to achieve better generalization and adaptation. To this end, SPARTAN modifies the standard Transformer attention mechanism by replacing it with a learnable, sparse "hard attention" mask. This mask, representing a state-dependent local causal graph, is learned by applying a sparsity regularizer on the connections between object-centric tokens. The authors evaluate SPARTAN on simulated physics environments and a real-world traffic dataset, showing that it learns more accurate causal graphs, which in turn leads to superior robustness against distractors and more efficient adaptation compared to standard Transformer baselines.

**Questions:**

Please refer to the weaknesses section above.

**Ethical Concerns:**

["NO or VERY MINOR ethics concerns only"]

**Final Justification:**

The rebuttal has addressed most of my concerns, so I will increase my score to 4.

**Quality:**

3

**Strengths And Weaknesses:**

**Strengths**

- The paper is built on the clear and intuitive hypothesis that physical interactions are best modeled as sparse, local causal events rather than through dense, fully-connected graphs. The paper is also very clearly written, with effective visualizations, and is easy to follow.

- The experimental results are thorough and convincing. SPARTAN not only matches or exceeds the prediction accuracy of dense Transformer baselines but also demonstrates a significantly better ability to identify the correct causal graph structure (lower SHD).

-  The core contribution—a learnable, sparse hard-attention mechanism within a Transformer—is a simple and direct implementation of the paper's central hypothesis.

**Weaknesses**

- The binary attention mechanism seems well suited to the contact-based interactions dominant in the chosen datasets. How do the authors see this approach extending to physical systems governed by continuous fields of influence, like gravity, where interactions are continuous and never truly zero?

- The matrix $\bar{A}$ treats all paths equally in the sparsity calculation. This seems to conflate direct influence with indirect, multi-hop influence. Have the authors considered alternative regularization schemes that might be more sensitive to the structure of these paths? For example, one could penalize direct connections in the final layers more heavily than longer, multi-hop paths that are resolved in earlier layers.

- The entire framework relies on the availability of high-quality, pre-disentangled object representations. In the simulated domains, this is achieved using ground-truth object masks to train a VAE. This is a significant assumption, as the performance of SPARTAN in a real-world setting would be fundamentally capped by the performance of an upstream object discovery and segmentation model, which is itself a challenging open problem.

- (Minor) Images are not referred to in the text.

- (Minor) Figure 2 is hard to parse. It would be nice to have a detailed subsection explaining it in text or in the caption.

---

> ### Author Rebuttal · Authors · 2025-07-30
>
> We thank reviewer gSAG for their comments. The reviewer has raised three main questions which we address below.
>
> **Extending to physics**: We first highlight that our model is able to capture non-contact effects, as evidenced in the Waymo dataset experiments, where interactions are mostly long-range. To answer the question more generally, we clarify that the proposed method is not just a binary attention, but rather a binary gate applied to the normal softmax attention. In the limit where the entities are fully connected, our architecture is equivalent to the standard transformer. Hence, we expect our model to predict as well as any other transformer-based architecture, albeit not learning sparse structures (because there is no sparse structure to be learned). A more nuanced point is that even when entities have non-zero influence on each other, there is still a tradeoff between interpretability and accuracy, and it is often still insightful to model emergent structures at the cost of negligible accuracy loss. In the gravity example, it is natural to think of a solar system as an independent system even though strictly speaking other solar systems have non-zero influences on its planets. In fact, all physical entities affect each other via gravity and other long-range forces, yet it is still useful to think of objects as separate entities that only interact when they make contact. Our lagrangian optimisation scheme (see App. A.2) formulates this problem as ‘finding the sparsest model that achieves a pre-determined error level’, which offers a flexible and practical algorithm for balancing this trade-off between accuracy and sparse structures.
>
> **Weighting paths equally**: The key desiderata for the sparsity measure is that it needs to track the existence of information flow from one token to another. In this perspective, we need to measure whether two tokens are connected at all and we do not necessarily care about whether the connections are multi-hop or direct. This is the main motivation for our path-counting matrix: we want to drive as many entries to zero as possible. It could be true that weighting different types of paths differently could offer practical benefits in terms of the optimisation process. However, for the experiments considered in this paper, the path-counting matrix was sufficient to induce good graph structures. We leave the investigation of whether other more involved regularisation schemes are useful to future work.
>
> **Requiring object masks**: Discovering and segmenting objects from images remains an active field of research that is orthogonal to our contribution. Since our approach can be readily applied to any pretrained representations such as slot-attention object embeddings, we expect the capability of our method to benefit from advances in this field. We clarify that, in our experiments, we opted for using ground-truth masked object representations primarily because it enables evaluation of the learned graphs: quantitatively evaluating the SHD of learned graphs requires mapping each latent embedding to the corresponding ground-truth object. To demonstrate the generality of our results, we have trained our model on an unsupervised learning based slot representation for the pong environment. Here, we pick a permutation that maps latent slots to ground-truth objects that results in the least SHD for reporting. The results are consistent with the main results in the paper: our model can achieve the same level of prediction while capturing the underlying graph structure more accurately (see below).
>
> | | SPARTAN | Transformer |
> |-|-|-|
> | Prediction error ($\pm$ SE) | 7.21 $\pm$ 0.69 | 6.90 $\pm$ 0.83 |
> | SHD  ($\pm$ SE) | **2.69** $\pm$ 0.02 | 8.52 $\pm$ 0.02 |
>
> We also thank the reviewer for the other minor suggestions and will fix these in an updated version of the paper.

---

> > ### Comment · Reviewer_gSAB · 2025-08-06
> >
> > The rebuttal has addressed most of my concerns, so I will increase my score to 4.

---

### Official Review · Reviewer_7Ngu · 2025-06-16

**Clarity:** 3
**Significance:** 3
**Originality:** 2
**Rating:** 4
**Confidence:** 4

**Summary:**

SPARTAN is a transformer-based method for learning sparse interaction patterns among entities, with a role of being a world model for RL agents.

**Questions:**

The ideas of this work (regarding sparsity, generalization, etc) are closely related to [1] and its follow-up [2], which are early works by Bengio's group, where sparse local interactions among entities are highlighted and then captured with transformer-based models. This paper would strongly benefit from acknowleging the commonalities and the differences from that line of work.


The idea on simple local models can be traced back to [3], which should be discussed properly.

[1] The Consciousness Prior
[2] A Consciousness-Inspired Planning Agent for Model-Based Reinforcement Learning
[3] Simple Local Models for Complex Dynamical Systems

**Ethical Concerns:**

["NO or VERY MINOR ethics concerns only"]

**Final Justification:**

The authors did not significantly improve the submission, despite having clarified one of my misunderstandings

**Limitations:**

The requirement for knowing the environment index I is not good for generalization. Can this be relaxed?

**Quality:**

3

**Strengths And Weaknesses:**

I find the approach to be simple and frankly quite amazed about the empirical performance of the Bernoulli binary masks.

I commend the authors for the use of performance metrics other than task returns. These make the experiments much more convincing.

---

> ### Author Rebuttal · Authors · 2025-07-30
>
> We thank reviewer 7Ngu for their comments and are delighted that the reviewer finds our experiments convincing. The reviewer raised two main questions, one concerning the connections to prior works, the other concerning the use of the environment index. Below we address these concerns.
>
> **Related works**: We agree that the consciousness prior is similar to our framework on a conceptual level. However, we note the key difference between our work and this line of work: we consider different kinds of sparsity. In the consciousness prior framework, the model is sparse in the sense of picking out a small subset of objects that are ‘active’ while leaving the other objects stationary. In contrast, in our work we model the dynamics of all objects, but use sparse attention to determine how each object depends on other objects. We thank the reviewer for the suggestion and will add a more detailed discussion in the updated text. We will also add a discussion on simple local models to add context to the current work.
>
> **Environment index**: We clarify that the environment index for the intervention token is only designed for cases where the model is trained on multi-environment datasets, but is not required for training our model. In the waymo dataset, for example, we do not have access to the environment index and therefore omit it. The use of the environment embedding (intervention token) is primarily for interpreting which object has changed its behaviour in each environment. This is akin to identifying the intervention targets in the causal discovery literature.

---

> ### Comment · Reviewer_7Ngu · 2025-08-03
>
> Thanks for the clarification of the environment index. Please try adding some more explicit explanations regarding this point.
>
> My own take of the "consciousness-prior" framework is that it is about the sparsity, not the stationarity, which may be an implicit assumption you made. The non-"active" entities can be also handled by system-1.

---

### Official Review · Reviewer_hr2c · 2025-06-29

**Clarity:** 3
**Significance:** 2
**Originality:** 1
**Rating:** 5
**Confidence:** 2

**Summary:**

The paper proposes a method to learn structured world models to recover the underlying casual structure of the environment. For this, the latent embeddings can be factored into independent causal mechanisms where with the data distribution shifts the changes to the causal structure is sparse. The paper presents a method to extend frameworks for state-based inputs to image observations and with different numbers of objects across scenes. The method, called SPARTAN, uses a multi-layer transformer where the adjacency matrix is sampled by sampling from Bernoulli distribution (to induce hard attention). The experiments on a few environments show that SPARTAN learns sparse graphs and is robust to change in dynamics.

**Questions:**

1. For the Waymo open dataset (Traffic), what does the observation include? Does it include the position of all objects or is the observation pixel-based?
2. What is the effect of environment index I during training (as discussed in Sec 3.2)? Is there any ablation on training without that information?
3. What is the value of $\lambda_s$ defined in Eq 6? How was this hyperparameter tuned and what is the effect of different values of this coefficient?
4. What is the effect of number of layers L? This looks like an important parameter as it also affects the adjacency matrix term for loss in Eq 6.

Typos
- Line 160– ‘the’ is mentioned two times.
- The legend in Figure 4 shows STRAW (for blue).

**Ethical Concerns:**

["NO or VERY MINOR ethics concerns only"]

**Final Justification:**

I am not an expert in this domain, but the authors have addressed my concerns, and I have no major concerns with this work.

**Limitations:**

The paper has discussed the limitations in Section 6.

**Quality:**

2

**Strengths And Weaknesses:**

### Strengths
1. The experiments support the claims proposed in the paper of using sparsity to accurately recover causal structure of the next state.
2. Overall the paper is well written and easy to follow.

### Weaknesses
1. A pseudocode of the training and inference steps would help better understand the algorithm. Also, this would highlight certain differences like the environment index is passed during training whereas the adaption token is obtained at test-time.
2. The proposed idea is pretty simple and uses a Bernoulli distribution to induce sparsity and hard attention. There is no theoretical justification for why the model should work. Although the papers mentions this in limitations but it would be good to have some analysis.

---

> ### Author Rebuttal · Authors · 2025-07-30
>
> We thank reviewer hr2c for their thoughtful comments and suggestions. Below we address the concerns and questions raised.
>
> **Weakness 1 - Pseudo code**: We update the paper with the relevant pseudocode for training and adaptation for better clarity.
>
> **Weakness 2 - Theoretical analysis**: In terms of theory, our work is grounded in the theoretical results developed in local causal discovery. The FCDL framework [1], for example, establishes theoretical results showing that sparsity regularisation will converge to the correct local causal graphs. These can be seen as theoretical justification of our approach, which provides a scalable method to perform sparsity regularisation on transformers. What these theories do not cover, however, is the case where the variables can change across scenes (as in the Waymo dataset). Establishing a rigorous theoretical framework for this case would require significant extension of current theory. This is beyond the scope of our current submission. We will update the text to add a more thorough discussion on how our work links to prior theoretical results.
>
> We also highlight that one of our main technical contributions is the path-counting algorithm over adjacency matrices which efficiently tracks interactions between tokens across layers. To the best of our knowledge, ours is the first method that successfully applies sparse regularisation to transformers in the context of discovering local structures. This solves a significant scalability challenge in the field of local causal discovery, where prior approaches can only work on simple environments. (See app C for a more detailed discussion). In this light, we sincerely invite the reviewer to reconsider their assessment of the originality of our work.
>
> **Question 1 - Waymo dataset**: In the waymo open dataset, there are two kinds of observations: for vehicles, the observations are x,y positions and heading angles for one second, sampled at 10Hz; for the lane markings, the observations are x y points that form a line. Both of these kinds of observations are essentially lines. Following the MTR architecture [2], these lines are encoded via a learnable polyline encoder which serves as the ‘tokeniser’. We will expand our description of the environment in App. B.3. to better explain the setting.
>
> **Question 2 - Effect of Env index**: We clarify that the environment index is not a prerequisite for training the model. We note that the Waymo experiment, for example, does not use labelled environments. The intervention token is only designed for the case where the model has access to multi-environment data where the behaviour of objects can be drastically different. This is inspired by the notion of soft interventions from causality which tells us which object is behaving differently, thereby providing an extra level of interpretability and adaptability.
>
> **Question 3 - Tuning Lambda**: The lambda parameter controls the strength of the sparsity regularisation. A value that is too high can result in a model that makes poor predictions as it does not use any connection. In practice, we use a lagrangian optimisation scheme (detailed in App. A.2) which serves as a schedule that automatically tunes the value of lambda to ensure that the model performance does not degrade due to sparsity. We provide a more detailed explanation of the formulation as well as example training dynamics in App. A. 2.
>
> **Question 4 - Number of layers**: The number of layers, much like in the standard transformer architecture, controls the expressivity of the model. Tuning this hyperparameter is a balance of how much data and compute is available. In our experiments, we have picked the number of layers to match what we use for the standard transformer baseline to ensure the comparison is fair. We note that the adjacency matrix regularisation term counts the number of paths between tokens and effectively counts the edges at every layer. We highlight that, empirically, our method works for a wide range of number of layers from three layers (in Pong) up to 12 layers (in Waymo).
>
> [1] Hwang et al., 2024, Fine-Grained Causal Dynamics Learning with Quantization for Improving Robustness in Reinforcement Learning
>
> [2] Shi et al., 2023, Motion Transformer with Global Intention Localization and Local Movement Refinement

---

> > ### Comment · Reviewer_hr2c · 2025-08-04
> >
> > The rebuttal addresses most of my concerns, and I have increased my score. I hope the authors add the pseudo-code to explain the training/inference pipeline.

---

### Official Review · Reviewer_AuXs · 2025-07-03

**Clarity:** 3
**Significance:** 2
**Originality:** 2
**Rating:** 4
**Confidence:** 3

**Summary:**

The paper proposes a Transformer-based world model that learns time-dependent local causal structures among entities from object-centric visual observations. Building on the idea that attention mechanisms can capture causal dependencies, the authors introduce hard attention combined with sparsity regularization. The resulting model, SPARTAN, is capable of adapting to interventions in the environment, even when the targets of those interventions are unknown. The method is evaluated across three domains, including Interventional Pong, CREATE, and the Waymo Open Dataset, and shows improved performance over baselines in terms of predicting causal structure, robustness to non-causal changes, and few-shot adaptation to novel dynamics.

**Questions:**

Questions for the Authors
1. How does the method handle changing numbers of objects across time, such as vehicles entering or leaving the scene?
2. Can the model deal with noisy or imperfect object segmentations, including missing or extra entities?
3. Does the method require consistent object indexing over time (i.e., object tracking)?

**Ethical Concerns:**

["NO or VERY MINOR ethics concerns only"]

**Final Justification:**

The authors have addressed most of my concerns during the rebuttal. My main remaining concerns are that the main technical contribution is the path-counting regularization, and the method depends on existing approaches like SAVi to operate on raw visual observations.

**Limitations:**

yes

**Quality:**

3

**Strengths And Weaknesses:**

Strengths
1. The paper is clearly written and well motivated.
2. The examples in Sections 2.2 and 2.3 are intuitive and help clarify the rationale behind the modeling approach.
3. Using the Waymo traffic dataset is a strong choice, as it reflects realistic, complex dynamics not easily captured in simulation environments.
4. The visualizations in Figure 2 effectively illustrate the discovered local causal structure and demonstrate close alignment with ground-truth graphs.
5. The adaptation mechanism, achieved through gradient-based optimization of intervention tokens, is interesting and useful for generalizing to new interventions with limited data.

Weaknesses
1. The assumption of access to ground-truth object segmentation for training is quite strong. Prior work, such as SlotFormer, learns from raw visual input without relying on ground-truth masks or object factorization.
2. Details on the experimental setup are insufficient. For example, the number of training episodes and the exact meaning of the “Pred. Err.” metric are not thoroughly described.
3. The approach primarily builds on existing components, including Gumbel-softmax sampling for hard attention and sparsity regularization. While the combination is effective, the technical contribution may be limited.

---

> ### Author Rebuttal · Authors · 2025-07-30
>
> We thank reviewer AuXs for their thorough review. The reviewer raised several concerns which we address below.
>
> **Requiring object masks**: We would like to point out that many existing works, including SlotFormer [1], use pre-trained object-centric representations such as SAVi [2] which handles the segmentation and tracking problems. Our approach can be readily applied to any such pretrained representations. In our experiments, we opted for using ground-truth masked object representations for the principal reason that it enables evaluation of the learned graphs: quantitatively evaluating the SHD of learned graphs requires mapping each latent embedding to the corresponding ground-truth object. Learned object embeddings can make this mapping noisy and ambiguous due to imperfect segmentations. Since learning high-quality representation is not the main focus of this work, we use ground-truth masked object representations to ensure the fairness and validity of our evaluation. Nevertheless, we understand that more evaluations can strengthen our claims. To demonstrate the generality of our results, we have trained our model on an unsupervised learning slot representation for the pong environment. The results are consistent with the main results in the paper: our model can achieve the same level of prediction while capturing the underlying graph structure more accurately (see below).
> | | SPARTAN | Transformer |
> |-|-|-|
> | Prediction error ($\pm$ SE) | 7.21 $\pm$ 0.69 | 6.90 $\pm$ 0.83 |
> | SHD  ($\pm$ SE) | **2.69** $\pm$ 0.02 | 8.52 $\pm$ 0.02 |
>
> **Details missing**: The total number of training episodes are 10k, 30k and 10k for the pong, CREATE and waymo datasets respectively. The prediction error metric measures the L2 distance between the prediction and the embedded ground-truth in the latent space. We will update the paper to include all experimental details. We will also open source the training code and the data-generation scripts to ensure reproducibility of our results.
>
> **Limited technical contribution**: We highlight our main technical contribution: we propose a method to perform sparsity regularisation on transformers via a novel path-counting regulariser as well as masked hard attention. While sparsity regularisation has been proposed in prior works on local causal discovery, existing methods remain difficult to scale beyond simple environments (see, for example, our discussion in App. C). Here, our approach of using a path-counting matrix is a key enabler for applying such sparsity penalty on the transformer architecture which is ubiquitous across many domains. As such, our method solves a crucial scalability problem in the field of local structure learning. Importantly, we note that our method is architecturally similar to the standard transformer **by design**, so that it can be readily used as a slot-in replacement for transformers.
>
> The reviewer also raised three questions regarding the practical aspects of our method.
> 1. In the waymo dataset, this is done by preprocessing the data to ensure no new vehicles enter the scene during the prediction horizon. Vehicles that no longer exist after a timestep are assigned a special token.
> 2. The model can operate on unsupervised learning based slot representations that are noisy and imperfect. Please see the weakness section for a more detailed discussion and extra results.
> 3. The transition model does not explicitly deal with tracking and assumes consistent indexing. In practice, object segmentation models such as SAVi [2] can perform tracking jointly with object discovery.
>
> In general, we point out that these problems are open challenges in object-centric world modeling that are orthogonal to our main contribution, namely, improving robustness, interpretability and adaptability via learning local structures. We note that our method can be used as a slot-in replacement for the standard transformer which is highly versatile. As such, we expect that our core contributions can be readily applied to any future or existing methods that can handle these problems.
>
> [1] Wu et al. 2024, SlotFormer: Unsupervised Visual Dynamics Simulation with Object-Centric Models.
>
> [2] Kipf et al. 2022, Conditional Object-Centric Learning from Video

---

> > ### Comment · Reviewer_AuXs · 2025-08-07
> >
> > I would like to thank the authors for their detailed rebuttal. Most of my concerns have been addressed, and I have updated my evaluation to Weak Accept. My main remaining concerns are that the main technical contribution is the path-counting regularization, and the method depends on existing approaches like SAVi to operate on raw visual observations.

---

### Note · Authors · 2025-08-12

We thank all the reviewers for their comments and suggestions. We are delighted to see that the reviewers in general found our rebuttal helpful. We take this opportunity to summarise some of the main discussion points raised and our plan to improve the manuscript for a camera-ready version.

**Requirement on pre-disentangled slots**: Reviewers AuXs and gSAB raised concerns about our method's requirements on having access to ground-truth object masks. We clarify that this is primarily done to enable the evaluation of learnt causal graphs. During the rebuttal, we have included new results to show that our method also works on self-supervised representations. These results will be added to the paper.

**Related works**: Reviewer 7Ngu suggested the Consciousness Prior line of works as related work to our method. We note the conceptual similarities despite the methodological differences, and will update our paper with a detailed discussion on how our work relates to it.

**Clarifications**: Reviewer hr2c and 7Ngu raised questions regarding the use of the intervention tokens. We clarify that these are not required but can be incorporated where available. We will clarify this in the text. As suggested by reviewer hr2c, we will also include a pseudo code of our training algorithm to improve clarity.

---

### Decision · Program_Chairs · 2025-09-17

**Decision:**

Accept (poster)

**Comment:**

This paper presents a transformer-based world modeling approach that explicitly enforces sparse structure in the causal interactions between elements in visual scenes. The reviewers acknowledged several strengths of this work, including clear and well-written presentation and intuitive examples that effectively justify the importance of the addressed problem.

The reviewers initially raised a few concerns but the authors adequately addressed them. Some reviewers also questioned whether the technical contribution was limited given that the approach builds upon existing developed components. However, the integration and application of these components to enforce sparse causal structure in world modeling represents a meaningful advance to me.
The work demonstrates both novelty in its approach to sparse world modeling and methodological rigour in its execution. Therefore, I recommend acceptance.